# Efficient Self-Attention Model for Speech Recognition-Based Assistive Robots Control

**DOI:** 10.3390/s23136056

**Published:** 2023-06-30

**Authors:** Samuel Poirier, Ulysse Côté-Allard, François Routhier, Alexandre Campeau-Lecours

**Affiliations:** 1Université Laval, Quebec City, QC G1V 0A6, Canadaalexandre.campeau-lecours@gmc.ulaval.ca (A.C.-L.); 2Centre for Interdisciplinary Research in Rehabilitation and Social Integration, CIUSSS de la Capitale-Nationale, Quebec City, QC G1M 2S8, Canada; 3Department of Technology Systems, University of Oslo, 0313 Oslo, Norway

**Keywords:** speech recognition, assistive robots, robotic assistive arm, keyword spotting, deep learning, self-attention, transfer learning, speech command, human–machine interface

## Abstract

Assistive robots are tools that people living with upper body disabilities can leverage to autonomously perform Activities of Daily Living (ADL). Unfortunately, conventional control methods still rely on low-dimensional, easy-to-implement interfaces such as joysticks that tend to be unintuitive and cumbersome to use. In contrast, vocal commands may represent a viable and intuitive alternative. This work represents an important step toward providing a viable vocal interface for people living with upper limb disabilities by proposing a novel lightweight vocal command recognition system. The proposed model leverages the MobileNet2 architecture, augmenting it with a novel approach to the self-attention mechanism, achieving a new state-of-the-art performance for Keyword Spotting (KWS) on the Google Speech Commands Dataset (GSCD). Moreover, this work presents a new dataset, referred to as the French Speech Commands Dataset (FSCD), comprising 4963 vocal command utterances. Using the GSCD as the source, we used Transfer Learning (TL) to adapt the model to this cross-language task. TL has been shown to significantly improve the model performance on the FSCD. The viability of the proposed approach is further demonstrated through real-life control of a robotic arm by four healthy participants using both the proposed vocal interface and a joystick.

## 1. Introduction

People living with upper limb disabilities face important challenges to accomplish their Activities of Daily Living (ADL). Over the past years, assistive devices have been developed to improve the autonomy of their users [1,2,3]. Robotics, in conjunction with artificial intelligence, has been seen a key solution to improve the quality of life of people living with physical disabilities. Notably, it has been shown that robotic devices can help the users regain a part of their autonomy and facilitate the accomplishment of their daily tasks [4,5].

One of the principal issues that arises with the control of robotic arms comes from the large number of Degrees of Freedom (DoFs) that the control interface allows the user to manage with a limited number of physical inputs. To lessen the difficulty for the users, the different movements are often regrouped in sub-groups. To perform a task, the user has to switch between these different sub-control groups to select the desired command, resulting in a lot of time and effort spent [6,7].

Interface intuitiveness however remains the Achilles’ heel within the field of assistive robotics. From [8], a definition of an intuitive control for human–machine interaction is “a technical system is, in the context of a certain task, intuitively usable while the particular user is able to interact effectively, not consciously using previous knowledge.” In recent years, multiple control interfaces have been proposed such as virtual joysticks [9,10]. However, these interfaces fail to provide an intuitive experience, increasing the difficulty and the cognitive load of the users [11,12]. Additionally, their use is still cumbersome and requires constant calibrations; this is why joysticks and sip-and-puff devices are still the predominantly used interfaces to this day [13].

In search of a simpler and more straightforward solution, speech command has been seen as one of the most promising control modes [14,15]. Nevertheless, speech recognition is a vast and complex field. As a matter of fact, the tone of the voice, speech rate and pronunciation of words are all characteristics that vary from one person to another. Historically, speech command interfaces have faced important performance issues due to the limited computational power and the type of algorithms used [16]. The advent of deep learning paired with high computing capabilities substantially improved the performance of speech control interfaces to the point of now being ubiquitous. Primary examples of said devices are smartphones, Internet of Things (IoT) and home automation [15,17]. These applications, however, use an internet connection to process the speech and produce word recognition. This requirement greatly reduces the applicability of speech recognition devices for assistive technologies as they should operate without any exterior dependency due to safety and autonomy concerns [18,19]. Also, the use of the internet for processing the data can pose some privacy issues for the users.

The general aim of this paper is to enable people living with upper limb disabilities to control assistive robotic arms intuitively in order to increase their autonomy. The objectives are (1) to develop a lightweight speaker independent Keyword Spotting (KWS) machine learning-based model for the control of assistive robotic arms with English words and compare it to state-of-the-art algorithms, (2) to leverage a Transfer Learning (TL) algorithm to enable robotic arm control with French voice commands (French being the native tongue of the target users) and to quantify the performance of the TL and (3) to assess the intuitiveness of the control interface for a robotic assistive arm.

This work’s main contribution is a novel and lightweight offline speaker-independent KWS model which, to the best of the authors knowledge, performs better than any known model (with fewer parameters). This increase in performance is achieved through the introduction of a novel segmented 2D self-attention module. Another contribution of this work is the introduction of a novel KWS dataset titled French Speech Commands Dataset (FSCD) comprising 4963 vocal command utterances from 33 participants (18 women and 15 men). Finally, another contribution of this work is to demonstrate the successful application of TL from the Google Speech Commands Dataset (GSCD) to the FSCD to achieves a vocal interface in French which allows four participants to control a robotic arm in real-time more intuitively than when using a joystick.

This paper is organized as follows. An overview of the related work in speech command recognition through machine learning algorithms and TL is given in Section 2. Section 3 presents two datasets: the Google Speech Commands Dataset (GSCD) and the FSCD. A presentation of the methods followed for the development of the proposed model as well as the audio features, the training setup and the comparison with the state-of-the-art KWS models is given in Section 4. Section 5 describes the implementation consideration of the algorithm for assistive robotic arms and the adaptation to the French vocal commands via TL. Section 6 presents the evaluation of the control interface for a robotic assistive arm with two experiments. Section 7 presents the results of the comparison between our model and the state-of-the-art KWS models and the result of the TL. Finally, the results are discussed in Section 8.

## 2. Related Work

Studies on voice-controlled interfaces using short commands for people living with disabilities have been carried out for over 30 years [16]. Different solutions leveraging the user-voice-based interaction with computers and robotic devices have been proposed. For instance, an interface designed for human–computer interaction used a vocal joystick based on discrete vocal commands paired with continuous vocal attributes (e.g., pitch and loudness) that are mapped to continuous control parameters [20]. Despite the improvements that this “vocal joystick” implemented compared to previous works, it still produces too many false positives. It is sensitive to noise and needs a specific training on the voice of the user to increase its performance [21]. This solution was then further developed to allow people with motor impairments to draw freely on a computer with only the use of their voice [22], but suffered from the same shortcomings as the vocal joystick. Other works extended the vocal joystick, such as [23], which used the non-verbal vocalization as a control mode for the Lynx 6 arm. In [24], a speaker-dependent algorithm based on Dynamic Time Warping (DTW) and K-Nearest Neighbor (KNN) grants the user the ability to control a custom-designed robot. Given the simplicity of this algorithm, its accuracy is highly dependent on the intonation and speech rate of the user as well as the surrounding noise. Furthermore, like all the previous algorithms, the user must record and train the system on his own voice to be able to use it (i.e., speaker-dependent). In [25], a robotic arm mounted on a wheelchair uses a vision system and a speech command interface where the user can give low-level commands (e.g., hand forward, gripper open) to control the end effector directly, and also high-level commands such as “pour in a drink”. This interface does not focus on the speech recognition algorithm and opts for the use of a now outdated generic speech recognition system, which brings performance limitations. A speech interface has been implemented for rehabilitation purposes as well in [26] where the users can control various movements of an exoskeleton robot with their voice. This interface has an efficient de-noising step based on wavelet for white noise, but it also faces some drawbacks, mainly due to the speaker-dependent system and the DTW algorithm, which requires words to be enunciated with the same intonation and speech rate as the training utterances. Other researchers [27,28] developed systems to control wheelchairs with voice commands. In each case, the speech recognition algorithm requires two models to be defined: the acoustic model and the language model. The use of two different models makes the training more complex and fragments the information that could be leveraged by a unified model to reach higher performance. Also, for these systems to reach a good performance, the users are required to update the acoustic model by producing plenty of dictation, correction and training data.

KWS is one of the most efficient methods used to implement a voice-based command interface that performs well with robotic arms [16]. KWS consists in the recognition of specific words in a context where multiple noises and words can overlap in an audio stream. The first methods [29,30,31] that allowed a high word recognition rate were based on the Hidden Markov Model (HMM). In this generative approach, a distinct HMM was trained for each word of the dictionary, and another HMM was added to handle the audio segments outside of the keywords dictionary. While performing inference, these HMMs use the Viterbi decoding algorithm, which make the real time predictions computationally expensive.

The paradigm then shifted from generative models to discriminative deep learning models, which yield higher performance and have lower computation cost. Several studies leveraged the highly performing deep learning models from the image classification field and transposed them to the speech recognition task. A highly accurate model used feed forward neural network [32] combined with a Gaussian mixture model (GMM), and largely surpassed the previously state-of-the-art HMMs. Another research [33] successfully demonstrated the high prediction accuracy and the possibility to maintain this accuracy while reducing the number of parameters of all the major baseline models (e.g., Deep Neural Network, Convolutional Neural Network, Long Short-Term Memory, Gated Recurrent Unit) of the deep learning universe. Recent research effort [34,35,36] have focused on the development of highly performing networks with an optimized amount of parameters.

The development of high performance KWS models led to the use of state-of-the-art models (e.g., Residual Neural Network) and the integration of attention mechanisms [37,38,39]. In [34], the application of residual layers to KWS outperformed previous state-of-the-art models and also allowed for a more compact architecture. Two of the models from this article are used for comparison: the res15, which is the most performant model and its lightweight counterpart res8-narrow. More recently, an architecture leveraging both residual connection and graph convolutional network has been proposed in [36]. From this article, the one with the best performance (the CENet-GCN-40) and the one with the smallest number of parameters (CENet-6) are also compared to our proposed model.

Since the publication of [40], self-attention mechanisms have been widely used in conjunction with every existing models to increase prediction accuracy and take advantage of its feed forward structure, which offers a low computational time. In the speech recognition sphere, large vocabulary models have leveraged self-attention for end-to-end training [41,42,43]. However, these models are not optimized for KWS tasks due to the high number of parameters that need to be trained to perform well. In [44], self-attention has been added to the extensively used Time Delay Neural Network (TDNN) and achieved the best performance for the least number of parameters found in the scientific literature. For the comparison process, the model with the best performance (tdnn-swsa) and the smallest one (swsa) were selected.

An important drawback of deep learning is its need for a large volume of data even when the model is designed to have a low number of trainable parameters. Since no large dataset is available for the required French voice commands, and creating a large dataset from scratch is expensive in terms of both time and money, TL is seen as a good alternative. TL is a useful method to enhance a model’s performance in a target task where data are scarce by leveraging a trained model in a data-rich related source domain. From the multiple ways to apply TL effectively for the speech processing [45], the performant TL method for deep neural networks described in [46,47] is used for the transfer from a large English speech command dataset to the smaller French dataset. With this technique, the KWS model is first trained on the source language (English). Then, the entire pre-trained model (or subset thereof) is retrained on the target language (French). This fine-tuning approach tends to yield better models than when simply using random initialization [46].

## 3. Speech Command Datasets

Within this work, two datasets are considered: the GSCD [48] and the novel FSCD which was created by the authors. The GSCD which is available at http://download.tensorflow.org/data/speech_commands_v0.01.tar.gz (accessed on 10 March 2020) served two purposes, namely: (1) to validate the proposed model and compare it to the current state-of-the-art models for the KWS task, (2) to pre-train the proposed model on prior to applying the TL algorithm toward the FSCD. Note that in this work, the first version of the GSCD is employed for comparison purposes with the different state-of-the-art models, and because it is the closest to FCSD. The GSDC consists of a vocabulary of 30 words for a total of 64,727 utterances (i.e., recorded examples of a voice command). The dataset is unbalanced in terms of number of utterances per class (e.g., the “stop” command has 2380 utterances whereas “bed” has 1713 utterances). Every utterance has a length of 1 s or less, recorded with a 16 kHz sample rate and is encoded as linear 16-bit single-channel PCM values. There is a total of 1881 different speakers, with a different number of utterances per speaker. Each speaker is identified via a unique identifier. In doing so, all utterances from a given speaker are contained within the same set (i.e., training, validation or test set). Ten of the 30 words are the actual commands that are used to control the assistive robotic arm in the experiments, the rest being filler words to help the model learn a wide range of phonemes. The words composing this dataset are presented in Table 1.

The FSCD is made readily available at the following link: https://arweave.net/rIRaVzZqR7aRa0U3gg2Hqxgpat0-wL1XFMZVUd0pq9M, accessed on 4 May 2023. This dataset consists of 4963 different vocal command samples from 33 participants (15 males, 18 females) aged between 20 and 45 years old. The FSCD is balanced: the standard deviation of the number of utterances per class being less than 4%. Every utterance has a length of 1 s, recorded with a 16 kHz sample rate and is encoded as linear 16-bit single-channel PCM values. The vocabulary of the dataset is made of 12 commands with a total of 29 words which are presented in Table 1.

A challenge of speech recognition is robustness to background noises and silence. Therefore, when training the model, several minute-long utterances of such signals were also used in the training of the model. From the given background, noise files of the GSCD, 3500 one-second-long silences and noise utterances were randomly sampled. These data were added to both datasets for the training.

## 4. Methods

### 4.1. Model Development

Inspired from the highly performant MobileNetV2 [49], the architecture of the proposed model (which can be seen in Figure 1) was designed to be compact and to ensure its efficiency on embedded devices. Five types of blocks compose the model: (1) the initial block, (2) the inverted residual block, (3) the transition block, (4) the Segmented 2D Self-Attention block and (5) the prediction block. The channel count and depth of the network are kept low to avoid overfitting and to reach our goal of a lightweight model. We proposed two models: the Segmented-SA-11k, which displays the best performance, and its smaller version, the Segmented-SA-7k.

#### 4.1.1. Initial Block

The purpose of this block is to increase the channel dimension from 1 to 8 before a first inverted residual block is applied. The use of this block (as opposed to directly using an inverted residual block) increased the performance of the network at an average of 0.3% for the validation set from ten different trainings. The initial block is composed of a bias-free 2D convolution layer with a 3 × 3 kernel and a stride of 1 followed by a Batch Normalization layer and a ReLU6 used as an activation function.

#### 4.1.2. Inverted Residual Block

The inverted residual block proposed in [49] is used as the main feature extractor of the network. This block allows an effective back-propagation of the gradient to the lower layers due to its residual connection between layers with a low number of channels (known as bottleneck layers). The network is composed of three inverted residual blocks. We added dilation with a rate of 2 since it was experimentally found to increase network performance without increasing model complexity.

#### 4.1.3. Transition Block

The first two inverted residual blocks are followed by a transition block. This block’s function is twofold: (1) to halve the frequency and time dimensions, reducing slowly the dimensionality of the input and (2) to increase the channel’s dimension to capture a greater share of meaningful features. Max pooling followed by a convolutional layer could have been used in place of this block to perform the down-sample and channel increase, but it was found experimentally that a large portion of the information would then be lost. This block is formed from a bias-free 2D convolution layer with a 3 × 3 kernel and a stride of 2, a Batch Normalization layer and a ReLU6 used as an activation function.

#### 4.1.4. Segmented 2D Self-Attention

Segmented 2D Self- Attention is a novel idea that extends the regular self-attention mechanism to the KWS task. The reasoning behind this approach is that non-adjacent phonemes do not have any relation with one another. Thus, the idea of the Segmented 2D Self-Attention is to restrict the self-attention to a 2D local space of the feature maps extracted by the network. In such a way, the network only has access to the local features around a phoneme and therefore, it must learn to extract important features directly in the vicinity of a phoneme.

The mechanism of the Segmented 2D Self-Attention is added after the third inverted residual block. This block was specifically placed at this position to leverage all the information extracted by the previous layers and to reduce the number of multiplications that the mechanism requires. A schematic representation of the Segmented 2D Self-Attention is presented in Figure 2. The projection of the input to the query Q, key K and value V spaces is performed with 2D convolution with a kernel size of 1 × 1. The 2D space is described by *c* different feature maps composed of the frequency *f* and the time *t* axes. The projection reduces the number of channels in a bottleneck fashion [50] to $\frac{c}{n}$ trainable filters, where *n* is fixed to obtain an integer multiple of the channel size. The three projections are obtained by the convolution ⊗ of the output tensor $X\in R^{c\times f\times t}$ from the preceding layer with the tensors $\left\{W_{Q},W_{K},W_{V}\right\}\in R^{\frac{c}{n}\times 1\times 1}$ resulting in $\left\{Q,K,V\right\}\in R^{\frac{c}{n}\times f\times t}$.
(1)$$Q=W_{Q}⊗X,$$
(2)$$K=W_{K}⊗X,$$
(3)$$V=W_{V}⊗X.$$

By rearranging the tensors {Q,K,V} to $R^{\frac{c}{n}\times ft}$, the softmax σ of the matrix multiplication of $Q^{T}$ and K gives a spatial self-attention map of size $R^{ft\times ft}$, and the matrix multiplication of Q and $K^{T}$ results in a channel self-attention map of size $R^{\frac{c}{n}\times \frac{c}{n}}$. The output of the global self-attention mechanism for the spatial self-attention is presented in Equation (Equation 4) whereas the output for the channel self-attention is given by Equation (Equation 5):(4)$$S=V(\sigma \left(\frac{Q^{T}K}{\sqrt{ft}}\right)),$$
(5)$$C=\sigma (\frac{QK^{T}}{\sqrt{\frac{c}{n}}})V.$$

As noted in [40], the matrix multiplication $Q^{T}K$ and $QK^{T}$ tends to grow large in magnitude, which causes the gradient to vanish due to the softmax operation. To counter this effect, the matrix multiplications need to be scaled down by a factor of $\frac{1}{\sqrt{ft}}$ and $\frac{1}{\sqrt{\frac{c}{n}}}$ respectively.

With the reasoning that global self-attention carried out on a complete utterance does not allow the model to fully capture the local and more subtle variations in the spectrogram, the feature space $R^{\frac{c}{n}\times f\times t}$ is segmented in smaller local grids. For each element (:,i,j) of $Q\in R^{\frac{c}{n}\times f\times t}$, a local tensor ${\hat{Q}}_{:,i,j}$ is extracted with a size of w+1 in the frequency axis and h+1 in the time axis, all the channels being included. Each element $q_{i,j}$ of ${\hat{Q}}_{:,i,j}$ is a vector composed of the $\frac{c}{n}$ feature maps. Supposing that the width *w* and height *h* of the local map are odd numbers, each local tensor can be extracted with:(6)$${\hat{Q}}_{:,i,j}=\left\{q_{i-\frac{\left(w-1\right)}{2},j-\frac{\left(h-1\right)}{2}},\ldots ,q_{i,j},\ldots ,q_{i+\frac{\left(w-1\right)}{2},j+\frac{\left(h-1\right)}{2}}\right\},$$
(7)$${\hat{K}}_{:,i,j}=\left\{k_{i-\frac{\left(w-1\right)}{2},j-\frac{\left(h-1\right)}{2}},\ldots ,k_{i,j},\ldots ,k_{i+\frac{\left(w-1\right)}{2},j+\frac{\left(h-1\right)}{2}}\right\},$$
(8)$${\hat{V}}_{:,i,j}=\left\{v_{i-\frac{\left(w-1\right)}{2},j-\frac{\left(h-1\right)}{2}},\ldots ,v_{i,j},\ldots ,v_{i+\frac{\left(w-1\right)}{2},j+\frac{\left(h-1\right)}{2}}\right\}.$$

The width *w* and height *h* are parameters that need to be explored as well as the overlap between each two consecutive maps. This overlap can be modified by adjusting the stride $s_{f}$ on the frequency axis and $s_{t}$ on the time axis. With $s_{f},s_{t}>1$, a part of the spectrogram elements are skipped, reducing the number of extracted local tensors. The padding *p* is adapted so the whole spatial input map (f×t) is included in the segmentation process. The maximum number of extracted local tensors for the frequency and time axes is given by the following equations:(9)$$u=\frac{f+2p-(w-1)-1}{s_{f}}+1,$$
(10)$$y=\frac{t+2p-(h-1)-1}{s_{t}}+1.$$

There is a total of uy segmented grids of size $R^{\frac{c}{n}\times w\times h}$ obtained from this operation for $\hat{Q},\hat{K},\hat{V}$ where i∈{0,…,u−1},j∈{0,…,y−1}.

After rearranging each local tensor $\hat{Q},\hat{K},\hat{V}$ to $R^{\frac{c}{n}\times wh}$, the process of 2D local self-attention is applied to each grid *m* extracted. Equation (Equation 4) and Equation (Equation 5) can then be rewritten for the spatial local self-attention to Equation (Equation 11) and and the channel local self-attention Equation (Equation 12) respectively:(11)$$S_{:,m}={\hat{V}}_{:,m}\left(\sigma \left(\frac{{\hat{Q}}_{:,m}^{T}{\hat{K}}_{:,m}}{\sqrt{wh}}\right)\right),$$
(12)$$C_{:,m}=\sigma \left(\frac{{\hat{Q}}_{:,m}{\hat{K}}_{:,m}^{T}}{\sqrt{\frac{c}{n}}}\right){\hat{V}}_{:,m}.$$

The local output tensors $S_{:,m}$ and $C_{:,m}$ are also of shape $R^{\frac{c}{n}\times wh}$. Both local tensors are then summed:(13)$$G_{:,m}=S_{:,m}+C_{:,m}.$$

The concatenation of all the local output tensors $G_{:,m}$ is of shape $G\in R^{uy\times \frac{c}{n}\times wh}$. By applying a summation over the overlapping section of all the extracted local tensors and then by reshaping it, we obtain the original shape before the segmentation occurred $G\in R^{\frac{c}{n}\times f\times t}$.

The final output is generated by a convolution with a kernel $W_{O}$ of shape $R^{c\times 1\times 1}$, expanding the channel’s dimension to its initial shape while leaving the other dimensions intact and allowing the residual to be added:(14)$$O=W_{O}⊗G+X.$$

#### 4.1.5. Prediction Block

The network ends with the prediction block. This block is composed of a global average pooling [51] to collapse the output of the Segmented 2D Self-Attention to a vector composed of the average of each channel. This vector is then fed to a fully connected layer with a number of neurons corresponding to the number of classes to predict. A softmax is used as the activation function to output the posteriors of each vocal command.

### 4.2. Audio Features

As stated in [52], the amount of data needed grows exponentially with the variables (i.e., dimensionality). Since a waveform signal sampled at 16 kHz contains 16,000 dimensions for a one-second utterance, this causes a large data sparsity that a dataset like the GSCD cannot satisfy. To lessen the impact of said sparsity, other audio features have been investigated. To this end, some of the most popular characteristics for speech recognition with a low number of input dimensions—the Mel-frequency cepstral coefficients (MFCCs)—have been considered, as well as the spectrogram.

The MFCCs are widely used throughout speech recognition algorithms. A time window of 20 ms with a stride of 10 ms was used to extract a total of 40 MFCCs. To reduce the edge effects caused by the STFT (also known as spectral leakage), we used a Hamming window function. For a signal of one second padded at its extremities, this pre-processing resulted in 101 contiguous frames of 40 dimensions each.

The second audio feature used was the spectrogram. Since it has been shown that MFCCs are not robust against noises [53], a reasonable choice was to let the neural network learn relevant characteristics from a signal with less data pre-processing. To obtain the spectrogram features, the Short-time Fourier transform (STFT) is first computed with a time window of 20 ms and a stride of 10 ms to which a Hamming window function was applied. These parameters were determined empirically. Then, only the magnitude is kept since overlapping windows allow the signal to be reconstructed from the magnitude alone, the phase only having a minor impact on sound quality. For a one-second signal padded at its extremities, the dimensions of the resulting spectrogram are 161 frequency bins by 101 time frames.

### 4.3. Training Setup

The GSCD has been used to assess and compare the performance of the proposed approach. The objective of the model is to discriminate between the ten different commands (see Table 1), all the other words of the dataset (20) being integrated in a “filling words” class. Silence utterances (i.e., no word being spoken) are grouped in a “silence” class, making a total of 12 classes. The output with the highest value is designated as the predicted class.

The training dataset comprises 1503 speakers (corresponding to 51,094 utterances), while the validation and test sets both comprise 189 speakers (6798 and 6835 utterances, respectively). The training and validation sets are used in the training phase whereas the test set is used only once at the end to validate the final performances of the trained models.

The model architecture was implemented, trained and tested using Pytorch [54]. Two types of models have been compared: models from the literature and models from the authors. For the models from the literature, they were implemented following the training procedure described in their respective articles. For the models developed by the authors, they have been trained with the following procedure.

The training was first performed by minimizing the categorical cross entropy. The Adam algorithm [55] is employed for the network optimization (lr = 3 × 10${}^{-3}$) with a batch size of 32. To reduce overfitting, a weight decay of 1 × 10${}^{-2}$ was added, and early stopping (patience of 20 epochs) was performed. Additionally, a learning rate annealing was applied with a factor of 2 and a patience of 10. The convolutional layers were initialized with the Kaiming initialization [56], the linear layers were initialized with the Xavier initialization [57], and the BN layers were initialized with a constant value of one for the weights and zero for the bias.

### 4.4. Comparison with Other Works

The performance of the proposed approach has been compared to the state-of-the-art, lightweight models that reached the highest accuracy rate for the lowest amount of parameters found in the scientific literature. The following models were compared to the models proposed in this paper: the res15 and its lightweight counterpart res8-narrow from [34], the CENet-GCN-40 and CENet-6 from [36] and the tdnn-swsa and swsa from [44]. These models have been implemented following their respective articles. A baseline MobileNetV2 model (i.e., the Segmented-SA-11k without the Segmented 2D Self-Attention block) is also compared to the proposed approach.

### 4.5. Transfer Learning

The FSCD that the authors created corresponds to only 7.7% of the size of the GSCD. To overcome the data scarcity of our relatively small dataset, the transfer of the representation learned on the GSCD has been tested with different TL methods. We chose the Segmented-SA-11k model since it is a good trade-off between performance and footprint. A smaller model also limits the risk of overfitting on a small dataset.

Fine-tuning [58] is arguably the most prevalent transfer learning technique in deep learning [46,59,60] due to both its implementation simplicity and good performance. Consequently, this work employs fine-tuning to leverage the GSCD in order to improve the performance of the model on the FSCD. Fine-tuning consists in initializing a determined number of layers of the model that will be trained on the target dataset with the corresponding layers of the pre-trained model. Then, it is possible to retrain on the target dataset the pre-trained model completely, or only partially, by freezing a portion of the layers (i.e., the weights of these layers are not modified by the backpropagation).

Two TL methods and a baseline are to be compared. The baseline corresponds to a random initialization of the model trained on the target dataset (i.e., no TL applied). The first TL method is a complete copy of the pre-trained model without any layers being frozen during the retraining. The second method also uses a complete copy of the pre-trained model, but all the layers are frozen, except the classification layer (i.e., the fully connected layer). For both methods and the baseline, the number of output neurons of the fully connected layer is increased from 12 to 14 (since there are 12 commands to predict in French, plus silence and unknowns). Thus, the weights of the fully connected layer have to be reinitialized for all the tested methods.

As suggested in [61], a two-step procedure is employed to compare the different transfer learning approaches (including the baseline without TL). First, Friedman’s test ranks the approaches amongst each other. Then, Holm’s post hoc test is applied (*n* = 11) using the best ranked method, being the entire model retraining, as a comparison basis.

## 5. Implementation Consideration for Practical Use of the Algorithm

Since the targeted application of the KWS algorithm is the control of an assistive robotic arm, which is for the most part mounted on a wheelchair, an embedded device is necessary for practical use. The memory footprint and the number of operations should then be taken into account in the design and choice of that model.

The memory footprint and the number of operations are both critical when designing a model to be implemented on embedded systems. To reduce the size of a model further, the quantization of the network can be carried out and this has shown to minimally affect the performance [62]. Following [33], a *small* model for microcontrollers can be defined by a maximum memory size of 80 KB and performs 6 MOPS maximum whereas a *medium* model has a size between 80 and 200 KB and performs between 6 and 20 MOPS. With this classification, considering the quantization to 8 bits of our models, the Segmented-SA-7k takes 16 KB and performs 5.5 MOPS, putting it in the *small* category. For the Segmented-SA-11k with a size of 25 KB and performing 6.3 MOPS, this model is in the *small* category in regard to memory, and in the *medium* category in regard to the number of operations (although being very close to the limit of the *small* category). Thus, the memory aspect is not a concern since, memory-wise, both models could easily fit in a microcontroller. Considering that the number of operations required is only 15% greater for the Segmented-SA-11k than the Segmented-SA-7k for a performance gain of 3.3% (using the spectrogram for input features), the Segmented-SA-11k has been chosen as the model for the target task of KWS for French voice commands.

## 6. Evaluation of the Control Interface for a Robotic Assistive Arm

To assess the performance of the proposed vocal command interface, four able-bodied participants aged 22 to 35 were asked to perform two tasks. All participants gave written informed consent prior to experiment onset; this study was approved by the local ethics committee (CIUSSS-CN; project #2019-1603, RIS_2018-616). Note that, despite the small sample size considered, previous works have shown that small sample sizes are sufficient to document the usability of assistive devices [63,64]. Each task was performed using both a standard Jaco joystick and the vocal command interface. At the start of each task, participants were randomly assigned to the joystick or the vocal command interface. Note that the participants did not need to provide any training data for their voices (i.e., the model did not require retraining), making the proposed method fully user independent.

The experiments were performed with a Jaco assistive robotic arm and the experiment setup presented in Figure 3. Jaco is an assistive robotic arm produced by Kinova Robotics [2] designed to be installed on a motorized wheelchair and used by people living with upper-body mobility limitations. Controlled through the wheelchair’s drive control, the Jaco arm allows the user to manipulate objects in their surroundings. This enhanced autonomy allows the user to perform ADLs such as drinking, eating and picking up items [5]. The experiments are based on TEMPA [65] (Test d’Evaluation des Membres Supérieurs des Personnes Âgées), a performance-based evaluation developed to help occupational therapists efficiently and accurately assess the performance of people living with upper limb disabilities.

The joystick control interface, presented in Figure 4, is composed of a stick—which outputs forward, backward, left and right commands—and a button. The button is used to toggle between modes. When “Mode 1” is active, the four stick signals are used to control the arm’s horizontal translations (forward, backward, left, and right); in “Mode 2”, the same signals control the vertical translation and rotation about the effector axis; in “Mode 3”, the stick controls the two remaining rotations; and in “Mode 4”, the stick commands the fingers’ opening and closing.

The vocal command interface is composed of a microphone (Microsoft LifeChat LX-3000) and a press button. Two main operating modes make up the vocal command interface, “Translation” and “Rotation”, which can be activated at any time by voicing the desired mode. Different action commands, chosen with the help of two occupational therapists for their clarity and simplicity, are then added to each mode. In the “Translation” mode, the available vocabulary includes *up, down, forward, backward, left, right, open* and *close*; these commands allow the user to perform the three available translations, as well as opening/closing the fingers. In the “Rotation” mode, the available vocabulary is *up, down, forward, backward, left, right, plus, minus, open* and *close*; these commands perform the three available rotations of the arm, as well as the rotation of the effector. Commands to open and close the fingers are also accessible in this mode. Users can change between operating modes at any time by voicing the desired operating mode. The user selects a motion with a vocal command (e.g., *left*) and by pressing the button, they can then make the robot perform the selected motion. The user can release the button to stop the arm’s motion and then press it again to continue the selected motion. The vocal command interface continuously listens to the surrounding sounds picked up by the microphone. The trained model ignores noise and words outside of its dictionary, classifying them as unknown words. The last recognized command stays active as long as no other command is recognized. Note that the words were voiced in French by the participants but are presented here in English for the sake of clarity. A flowchart for both control interfaces logic is presented in Figure 5.

### 6.1. Task 1

The first experiment was the first task proposed in the TEMPA (i.e.,“Pick-up and move a jar”) and consisted of controlling the Jaco arm to reach, grasp, and lift a coffee jar placed on the top shelf and move it onto the middle of the bottom shelf. Each participant performed this task with both the joystick and the vocal command interface. At the start of the task, the Jaco arm was in its default home position, and the jar was positioned on the right side of the top shelf. Each participant repeated the experiment three times using each control interface.

### 6.2. Task 2

The second experiment required participants to perform the third task proposed by the TEMPA (i.e., “Pour water from a pitcher into a glass”). The participants controlled the Jaco arm in order to reach, grasp, and lift a bottle filled with water placed on the top shelf and pour water into a glass placed in the middle of the bottom shelf. The task was performed with both the vocal command interface and the joystick. At the start of the task, the Jaco arm was in its default home position and the bottle was positioned on the right side of the top shelf. This experiment was also repeated three times by each participant and with each control interface.

### 6.3. Subjective Evaluation

In order to compare the two interfaces, each participant completed a questionnaire consisting of seven questions. The questionnaire, based on [66], was designed to evaluate subjective measures related to the use of the interfaces and gain insights for future development of the vocal command. The questionnaire used a five-point Likert scale.

## 7. Results

### 7.1. Comparison with Other Works

The accuracy results of each model trained and tested on the GSCD are presented in Table 2. They were obtained from a test run on the standard test set provided by the GSCD of ten trainings from different initialization seeds. The prediction accuracy is used for the performance comparison of the models. The accuracy is the fraction of correct predictions over all the predictions made. The 95% confidence interval is computed from a t-distribution with nine degrees of freedom.

### 7.2. Result of Transfer Learning on the French Dataset

To assess the performance of the transfer learning from the GSCD (source) to the FSCD (target), the test dataset of the FSCD is used to compare the different TL methods at each epoch. Of the 33 participants who contributed vocal command utterances to the FSCD, 22 were randomly selected to be used as the training and validation sets while the remaining 11 formed the test set. The test set is only used to assess the performance of the TL. It does not serve the purpose of guiding the training of the model; the training and validation sets are used for this motive. In the same way as the GSCD, all utterances produced by the same speaker are only part of one set. The spectrogram was used for the input features. The results of the two TL methods and the baseline are presented in Figure 6. The score reported at each epoch is the average of ten trainings from different initialization seeds on the test dataset of the target.

### 7.3. Result of the Task Completion Time

The completion time was recorded for each test in both experiments. The results of the first experiment are shown in Figure 7. The average completion time using the joystick was 55.2 s with a standard deviation of 6.6 s, while the average completion time using the vocal control interface was 86.2 s with a standard deviation of 17.8 s.

The results of the second experiment are shown in Figure 7. The average completion time using the joystick was 91.5 s with a standard deviation of 20.1 s, while the average completion time using the vocal control interface was 109.2 s with a standard deviation of 18.1 s.

### 7.4. Control Interface Intuitiveness

The subjective evaluation conducted via the questionnaire was aimed at comparing the two interfaces and developing a better understanding of the strengths and weaknesses of the vocal command interface for future improvements. The Likert scale was scored from 1 (Totally Disagree) to 5 (Totally Agree), with 3 being the neutral option (Neither Agree Nor Disagree). The results of the questionnaire are presented in Figure 8.

All participants found the vocal control interface useful (Q2, average: 4.75) and easy to learn (Q7, average: 4.75). However, participants expressed that the vocal command interface did not improve their performance (Q3, average: 1.75), an observation that was consistent with the task completion time.

While it is a less striking result, participants agreed that the vocal command interface was easier to use than the joystick (Q4, average: 3.5); this is consistent with their observation that the joystick required more effort to use than the vocal command (Q1, average: 3.75). More notably, participants found that using the vocal control interface required less concentration than the joystick (Q5, average: 4); participants also found it to be more intuitive to use than the joystick (Q6, average: 4.25). One participant observed that she “didn’t have to think much” when using the vocal command interface. In fact, participants indicated that the vocal command interface design made it easy to navigate within the control modes simply by voicing the mode and the movement to execute.

## 8. Discussion

### 8.1. Model Performance

The initial step for the comparison with other works was to assess if our implementation of each model from their respective articles achieved an accuracy close to the reported results. Each implementation obtained a difference in accuracy of less than 0.2% from their original implementation reported in their respective article; hence, they were judged satisfactory.

For the MFCCs features, we can see from Table 2 that the proposed Segmented-SA-11k model achieves a superior classification accuracy to all the other models. More precisely, it shows a gain of 0.6% in accuracy (97.1% vs. 96.5%) from the second best model (i.e., CENet-GCN-40). According to non-parametric Wilcoxon signed-rank tests with paired data [61] where each person of the test dataset is considered as a single dataset (*n* = 188), the median of the performance of the Segmented-SA-11k is significantly different from the median of the CENet-GCN-40 (*p*-value < 0.000001). This performance is achieved with a reduction of more than a factor of six in model size (11.3 k vs. 72.3 k) between these two models. Compared to the model with the closest number of parameters (i.e., tdnn-swsa), it shows a performance boost of 1.4% with 0.7 k less parameters (11.3 k vs. 12 k). The smallest proposed model, the Segmented-SA-7k, which is also the one with the smallest number of parameters of all the tested models, yields a performance of 95.4%, which is close to the res15 (95.8%) even though it has more than 33 times fewer parameters. Compared to the model with the nearest number of parameters (SWSA), it shows a performance improvement of 5.1%.

For the spectrogram features for which the results are presented in the last column of Table 2, we note a constant gain in performance for the majority of the models. This result shows that a larger and less pre-processed input can improve the performance of a model with a high representation power. The only models presenting a lower performance with the spectrogram are also the ones with a low number of parameters (i.e SWSA and Segmented-SA-7k). Considering that the spectrogram is four times the size of the MFCCs in terms of frequency and that its features are not as uncorrelated as the MFCCs, this can prevent models with a limited representation from fully capturing all of the features of a larger input. Thus, a smaller and more pre-processed input such as the MFCCs can facilitate the learning of such models. The Segmented-SA-11k model still has the highest performance (97.4%), which is 0.4% more than the second-best model (CENet-GCN-40). The median accuracy of the Segmented-SA-11k is also significantly different from the median of the CENet-GCN-40 according to non-parametric Wilcoxon signed-rank tests with paired data where each person of the test dataset is considered as a single dataset (*p*-value < 0.000001).

Furthermore, the results showcased in Table 2 highlight the improvement obtained from utilizing the Segmented 2D Self-attention in the proposed models (Segmented-SA-7K and Segmented-SA-11K) compared to the baseline MobileNetV2 for both MFCCs and spectrogram features.

### 8.2. Transfer Learning Performance

From Figure 6, it can be observed that the impact of the transfer learning on the performance of the model on the FSCD is highly notable. Indeed, the performance of the model trained directly on the FSCD from a random initialization of all its layers yielded a poor performance (green dotted curve), with a maximum of 63.97% reached at the 58th epoch (average of 10 runs from different initialization seeds). This result demonstrated that the French Dataset does not contain enough data for the model to discriminate efficiently between the different speech commands.

Both methods where transfer learning was applied reached a good performance. The retraining of the entire model (no frozen weights) initialized from the pre-trained model on the source dataset yielded the best performance (94.78% as peak performance of the average of 10 runs at the 43rd epoch). The fine-tuning of the fully connected layer, while the rest of the network was frozen, led to a lower performance than the entire model retraining. Nevertheless, this method of transfer learning still reached a performance peak of 90.15% on the average of 10 runs at the 60th epoch. Since the backpropagation was performed only on the last layer of the network, the retraining took a little less than five minutes for 70 epochs. It is also to be noted that, with TL, the model was trained on a dataset 13 times smaller and still reached a classification score relatively close to the one obtained on the GSCD (97.4%).

The TL approach, which retrained the entire model, was shown to achieve better performance than the other two approaches, and this difference was judged significant following the two-step procedure described in section IV.E (*p* < 0.000001 in both cases).

### 8.3. Participants’ Evaluation

Both experiments showed that use of the vocal command interface resulted in slower task completion times compared to the standard Jaco joystick (average of 55.2 s vs. 86.2 s for task 1, and 91.5 s vs. 109.2 s for task 2). These longer completion times can be partially explained by speech processing delays. Although vocal command processing creates a short delay (75 milliseconds), the overall delay observed was mainly induced by the algorithm, which waits for the sound signal to be under the noise threshold for half a second before proceeding to the speech recognition.

However, although healthy participants were able to easily operate the joystick and quickly alternate between modes with the help of the push button, people living with upper-body incapacities may have difficulty operating the joystick (e.g., due to a lack of dexterity or the presence of spasms) or switching between the joystick control and the push button. Other users simply cannot operate a joystick (e.g., people with quadriplegia). Since the vocal command interface uses only a push button, which can be placed anywhere (e.g., a head push button for people with quadriplegia), the hypothesis is that for the target users, task completion times with the vocal control interface could be comparable or better than task completion times with the joystick control interface.

### 8.4. Experiment Limitations

Although these experiments served the purpose of testing the vocal command interface in real-life conditions, we remain cautious in relation to drawing conclusions about these results. While tests involving small samples (*n* = 4–5) have been shown to be sufficient to document the usability of assistive devices [63,64], this is still a very limited sample. An even more critical factor is that these experiments were not conducted with end users, but rather with healthy participants who had no physical limitations operating the standard Jaco joystick. Both experiments, however, provided good performance assessments of the vocal command interface in real conditions, as well as valuable insights concerning possible improvements. Another important factor to consider is noise level, which was relatively controlled in the test environment where the experiments were conducted. Vocal command interface performance is in fact limited in low signal-to-noise ratio environments.

## 9. Conclusions

This paper presents a lightweight speaker-independent speech recognition model for the KWS task that was shown to outperform the current state-of-the-art models. Moreover, this work presents the Segmented 2D Self-Attention mechanism, a self-attention mechanism designed to improve the performance of the KWS task. On the test set of the GSCD, the proposed model (Segmented-SA-11k) achieved 97.1% with the MFCCs as input features, which is 0.6% higher than the second best model, the CENet-GCN-40. With the spectrogram features, this model achieved 97.4%, a gain of 0.4% from the CENet-GCN-40. These performance gains were achieved despite the fact that the Segmented-SA-11k has a model size six times smaller compared to the CENet-GCN-40 (11.3 k vs. 72.3 k). The performance differences were shown to be significant for both input features and made the proposed model convenient to use in practice.

In addition, this work aimed at enabling robotic arm control with French voice commands (as French is the first language of the target users). TL methods have been used to overcome the data scarcity problem of the authors custom French dataset by transferring the learned representation of the network from the GSCD toward the FSCD. Both transfer learning methods that have been tested yielded a high performance: 94.78% when retraining the entire model (no frozen weights) and 90.15% for the fine-tuning of the fully connected layer, both results being the peak performance on the average of 10 runs. The random initialization and retraining of the entire network on the FSCD, which is equivalent to applying no TL, yielded a peak result of only 63.97% on the average of 10 runs. This result emphasized the necessity of using a TL approach when implementing a vocal interface from a small dataset. Moreover, the difference between the entire model retraining and the two other methods (fully connected layer fine tuning and baseline without TL) were shown to be significant.

Our vocal command interface was evaluated on two tasks from the TEMPA. Four able-bodied participants performed both tasks using the standard Jaco joystick and the vocal command interface. Importantly, the four participants did not provide any training data to the algorithm, and thus the proposed method was evaluated in a fully user-independent setting. Although participants required more time to complete both experimental tasks using the vocal command interface, they nonetheless expressed that the vocal command interface was more intuitive to use and that performing the tasks required less concentration than with the joystick. Participant perception of the vocal interface was also positive, as they reported that it was both useful and easy to learn.

Future works will focus on three main developments. Firstly, the robustness of the model to noisy environments and out of dictionary words will be increased by adding more examples to the training set. These examples could be gathered from different large publicly available datasets such as the Common Voice [67]. Secondly, the proposed model will be implemented on an embedded system for utility and mobility reasons. Finally, the algorithm will be tested with people living with upper limb disabilities to assess its real-use-case performance.

## Figures and Tables

**Figure 1 sensors-23-06056-f001:**
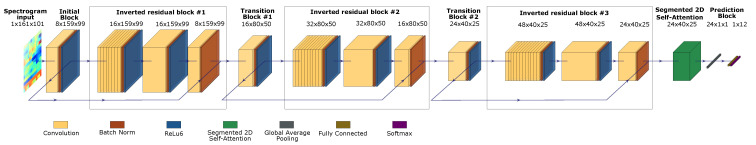
The proposed Segmented-SA-11k composed of an initial convolution block followed by three inverted residual blocks interspersed by transition block. The Segmented 2D Self-Attention block is added after the last inverted residual block. The number of output neurons in the prediction block corresponds to the 12 different vocal commands to predict from the Google Speech Commands Dataset (GSCD), including silence and unknown words.

**Figure 2 sensors-23-06056-f002:**
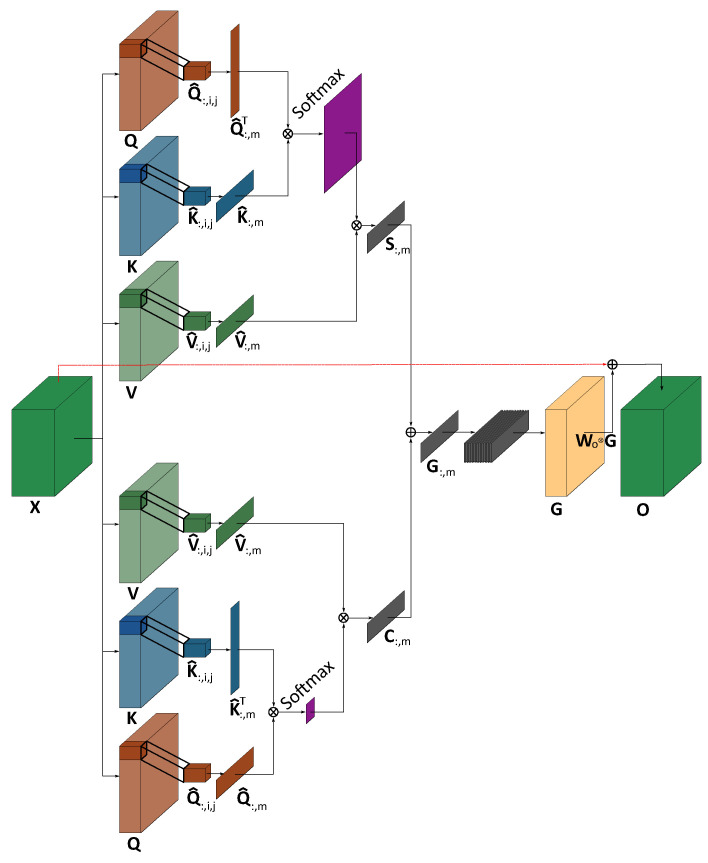
A schematic representation of the proposed Segmented 2D Self-Attention block where the upper part is the spatial local self-attention and the lower part describes the channel local self-attention. Both parts are then combined and the residual of the input is added.

**Figure 3 sensors-23-06056-f003:**
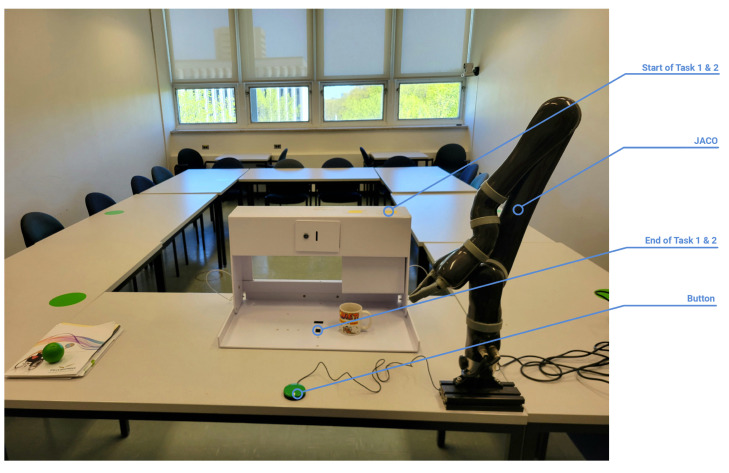
The experiment setup where two tasks proposed in the TEMPA were performed. The Jaco as well as the button used to switch mode when using the joystick or activate the movement of the Jaco when using the vocal command are presented. The starting and ending position of both tasks are also illustrated. Note that the glass used to receive the water in the second task is not in the Figure for the sake of clarity.

**Figure 4 sensors-23-06056-f004:**
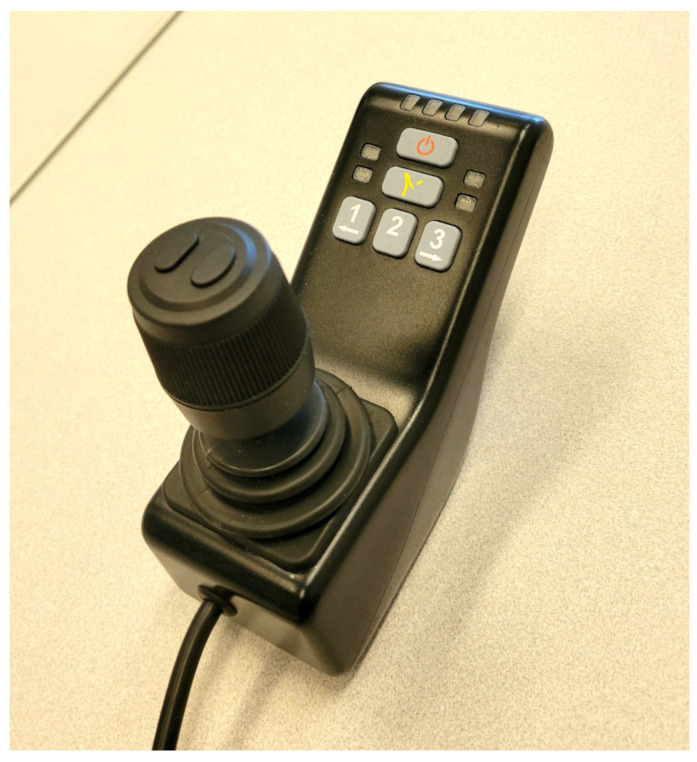
The standard Jaco joystick used as a reference point for performance comparison in assessing the qualitative and quantitative aspects of the proposed vocal command interface.

**Figure 5 sensors-23-06056-f005:**
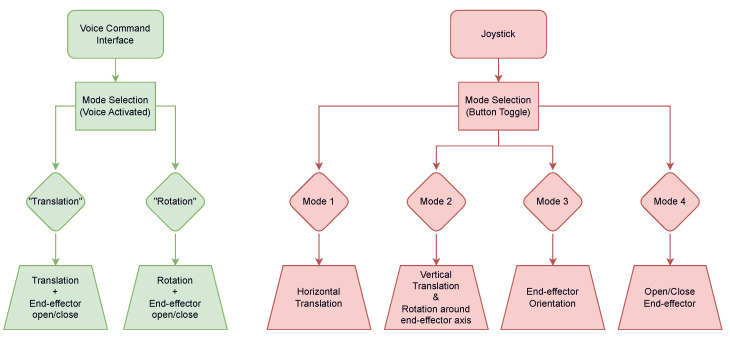
The flowchart detailing the process of both the vocal command interface and the joystick in order to control the Jaco movements.

**Figure 6 sensors-23-06056-f006:**
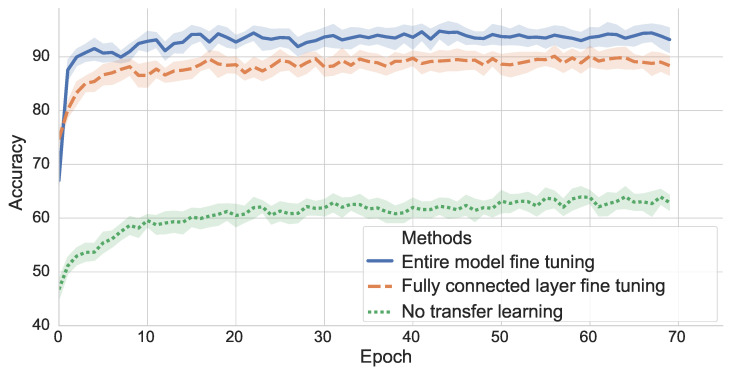
Results at each epoch on the test dataset of the French Speech Commands Dataset (FSCD) with the two TL methods and the baseline (without any TL).

**Figure 7 sensors-23-06056-f007:**
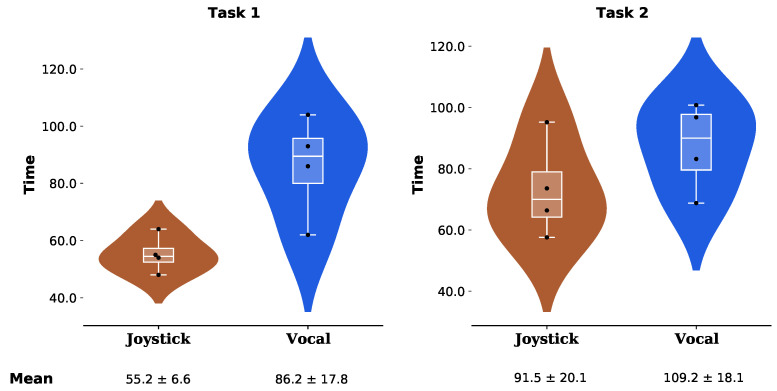
Probability density estimation of the experiments’ completion time using the standard Jaco joystick and the vocal command interface to control the Jaco robot arm for the first and second task.

**Figure 8 sensors-23-06056-f008:**
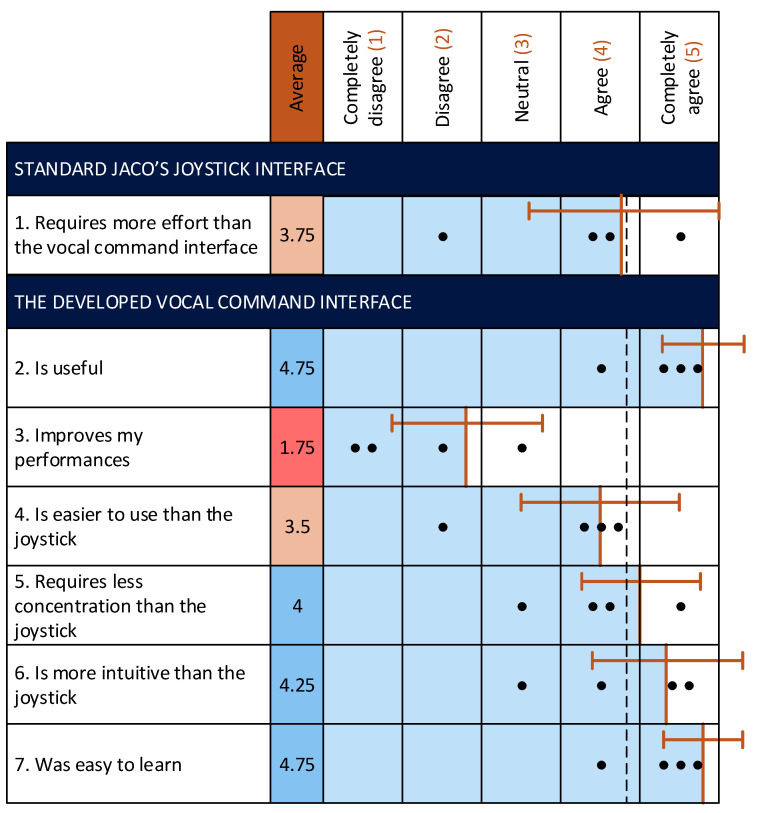
Results of the questionnaire completed by each participant after having performed both tasks to compare the standard Jaco joystick with the vocal command interface. The dots represent the score given by each participant.

**Table 1 sensors-23-06056-t001:** Commands and filling words of both datasets used for the training, validation and testing of the developed models.

Datasets	Commands	Filling Words
Google Speech Commands Dataset	“yes”, “no”, “up”, “down”, “left”, “right”, “on”, “off”, “stop”, “go”	“bed”, “bird”, “cat”, “dog”, “happy”, “house”, “marvin”, “sheila”, “tree”, “wow”, and the digits zero to nine.
French Speech Commands Dataset	“arrière”, “avant”, “descendre”, “droite”, “fermer”, “gauche”, “moins”, “monter”, “ouvrir”, “plus”, “rotation”, “translation”	“assiette”, “bouche”, “combiner”, “doigts”, “lentement”, “mode”, “position”, “précédent”, “rapide”, “sauver”, “suivant”, “tourner”, and the digits one to five.

**Table 2 sensors-23-06056-t002:** Test accuracy with MFCCs and spectrogram for input with 95% confidence intervals (on 10 different initialized trainings) and parameters count of each implemented models.

Model	Rank	# Param.	Accuracy on MFCC	Accuracy on Spectrogram
CENet-6	6	16.2 k	93.9% ± 0.24	94.7% ± 0.47
CENet-GCN-40	2	72.3 k	96.5% ± 0.19	97.0% ± 0.22
swsa	9	8 k	90.3% ± 0.18	88.9% ± 0.36
tdnn-swsa	4	12 k	95.7% ± 0.13	96.2% ± 0.29
res15	3	238 k	95.8% ± 0.22	96.4% ± 0.11
res8-narrow	8	19.9 k	90.2% ± 0.84	90.5% ± 0.45
Baseline MobileNetV2	7	9 k	92.1% ± 0.20	93.6% ± 0.33
Segmented-SA-7k	5	**7.1 k**	95.4% ± 0.23	95.0% ± 0.26
Segmented-SA-11k	**1**	11.3 k	**97.1% ± 0.16**	**97.4% ± 0.35**

## Data Availability

Publicly available datasets were analyzed in this study. This data can be found here for the GSCD: http://download.tensorflow.org/data/speech_commands_v0.01.tar.gz (accessed on 10 March 2020) and here for the FSCD: https://arweave.net/rIRaVzZqR7aRa0U3gg2Hqxgpat0-wL1XFMZVUd0pq9M (accessed on 10 March 2020).

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
