# Peer review of "Efficient Self-Attention Model for Speech Recognition-Based Assistive Robots Control"

_sensors, 2023, doi:10.3390/s23136056_

Round 1
Reviewer 1 Report
could you make a process diagram of the speech recognition model you present?
What types of robots, which ones would you implement your voice control model?
Why speech recognition? Why not go from speech to text and do the analysis through natural language processing, the cost is not less?
In command-based speech recognition, it does not matter the gender or age of the person?
Reviewer 2 Report
The meaning of self-attention is ambiguous in the title of the paper. Do the authors mean 'Self-attention' as 'voice or speech'? Where is Self-attention shown in the result table or graph of the paper? There is no data related to self-attention anywhere in the table or Figure 1 to Figure 5 presented in the paper.
Experimental design for robot control should be presented in a table. In addition, it is necessary to present the JACO joystick that brought the results of Figure 5 and the experimental scene comparing the voice interface.
Reviewer 3 Report
This paper presents a lightweight speaker-independent speech recognition model for the KWS (Keyword Spotting) task. The work is useful for robotic arm control. Some novel approaches in the model are presented for improving the accuracy of the speech recognition. In addition, transfer learning methods have been used to overcome the data scarcity problem of the authors custom French dataset. The following are some recommends for modification:
(1) “7.4 Control interface intuitiveness” ,please capitalize the first letter of each word and maintain consistent title format.
(2) Some figures of robots and experiment environments are needed for describing “Task 1” and “Task 2” in detail.
(3) Please analyze the experimental results in the figures such as Fig.3.
Minor editing of English language required
Reviewer 4 Report
This research is very interesting because it is very important for people with upper limb disabilities to control personal robots using voice to increase their autonomy.
The number of subjects who participated in the evaluation experiment of the control interface of the robot assist arm was four, but this number of participants seems to be too small from a statistical point of view. How about increasing the number of subjects to ensure reliability? Since healthy subjects are the target, isn't it possible to increase the number of subjects?
We expect that in the near future, you will conduct experiments using people with upper-limb function impairments as subjects, and confirm the effectiveness of the proposed method by comparing it with conventional methods that use speech command recognition.
Round 2
Reviewer 2 Report
The overall description of the paper is too complicated and there are many words that are not appropriate. For examples, in the introduction section, the terms are different every time, such as this project, this work, and this paper. Overall, in order to present it as the core of this paper, it is not well described that voice data from thirty three are matrix-calculated to perform robot work.
Learning with 33 people's data is too small. Why is Figure 3 presented without people?? A participant can sit (with his/her face covered). Also how is figure 4 also powered by vocal commands? Where does the sound go in the joystick in Fig.4? The probability density in Figure 7 is wrong and the sample distribution is correct.
